# Gaussian Quadrature for Kernel Features

**Tri Dao**
Department of Computer Science
Stanford University
Stanford, CA 94305
trid@stanford.edu

**Christopher De Sa**
Department of Computer Science
Cornell University
Ithaca, NY 14853
cdesa@cs.cornell.edu

**Christopher Ré**
Department of Computer Science
Stanford University
Stanford, CA 94305
chrismre@cs.stanford.edu

## Abstract

Kernel methods have recently attracted resurgent interest, showing performance competitive with deep neural networks in tasks such as speech recognition. The random Fourier features map is a technique commonly used to scale up kernel machines, but employing the randomized feature map means that $O(\epsilon^{-2})$ samples are required to achieve an approximation error of at most $\epsilon$. We investigate some alternative schemes for constructing feature maps that are deterministic, rather than random, by approximating the kernel in the frequency domain using Gaussian quadrature. We show that deterministic feature maps can be constructed, for any $\gamma > 0$, to achieve error $\epsilon$ with $O(e^{e^\gamma} + \epsilon^{-1/\gamma})$ samples as $\epsilon$ goes to 0. Our method works particularly well with sparse ANOVA kernels, which are inspired by the convolutional layer of CNNs. We validate our methods on datasets in different domains, such as MNIST and TIMIT, showing that deterministic features are faster to generate and achieve accuracy comparable to the state-of-the-art kernel methods based on random Fourier features.

## 1 Introduction

Kernel machines are frequently used to solve a wide variety of problems in machine learning [26]. They have gained resurgent interest and have recently been shown [13, 18, 21, 19, 22] to be competitive with deep neural networks in some tasks such as speech recognition on large datasets. A kernel machine is one that handles input $x_1, \ldots, x_n$, represented as vectors in $\mathbb{R}^d$, only in terms of some *kernel function* $k : \mathbb{R}^d \times \mathbb{R}^d \to \mathbb{R}$ of pairs of data points $k(x_i, x_j)$. This representation is attractive for classification problems because one can learn non-linear decision boundaries directly on the input without having to extract features before training a linear classifier.

One well-known downside of kernel machines is the fact that they scale poorly to large datasets. Naive kernel methods, which operate on the *Gram matrix* $G_{i,j} = k(x_i, x_j)$ of the data, can take a very long time to run because the Gram matrix itself requires $O(n^2)$ space and many operations on it (e.g., the singular value decomposition) take up to $O(n^3)$ time. Rahimi and Recht [23] proposed a solution to this problem: approximating the kernel with an inner product in a higher-dimensional space. Specifically, they suggest constructing a feature map $z : \mathbb{R}^d \to \mathbb{R}^D$ such that $k(x, y) \approx \langle z(x), z(y) \rangle$. This approximation enables kernel machines to use scalable linear methods for solving classification problems and to avoid the pitfalls of naive kernel methods by not materializing the Gram matrix.

In the case of shift-invariant kernels, one technique that was proposed for constructing the function $z$ is *random Fourier features* [23]. This data-independent method approximates the Fourier transform integral (1) of the kernel by averaging Monte-Carlo samples, which allows for arbitrarily-good estimates of the kernel function $k$. Rahimi and Recht [23] proved that if the feature map has dimension $D = \tilde{\Omega}\left(\frac{d}{\epsilon^2}\right)$ then, with constant probability, the approximation $\langle z(x), z(y) \rangle$ is uniformly $\epsilon$-close to the true kernel on a bounded set. While the random Fourier features method has proven to be effective in solving practical problems, it comes with some caveats. Most importantly, the accuracy guarantees are only probabilistic and there is no way to easily compute, for a particular random sample, whether the desired accuracy is achieved.

Our aim is to understand to what extent randomness is necessary to approximate a kernel. We thus propose a fundamentally different scheme for constructing the feature map $z$. While still approximating the kernel's Fourier transform integral (1) with a discrete sum, we select the sample points and weights *deterministically*. This gets around the issue of probabilistic-only guarantees by removing the randomness from the algorithm. For small dimension, deterministic maps yield significantly lower error. As the dimension increases, some random sampling may become necessary, and our theoretical insights provide a new approach to sampling. Moreover, for a particular class of kernels called sparse ANOVA kernels (also known as convolutional kernels as they are similar to the convolutional layer in CNNs) which have shown state-of-the-art performance in speech recognition [22], deterministic maps require fewer samples than random Fourier features, both in terms of the desired error and the kernel size. We make the following contributions:

- In Section 3, we describe how to deterministically construct a feature map $z$ for the class of subgaussian kernels (which can approximate any kernel well) that has exponentially small (in $D$) approximation error.
- In Section 4, for sparse ANOVA kernels, we show that our method produces good estimates using only $O(d)$ samples, whereas random Fourier features requires $O(d^3)$ samples.
- In Section 5, we validate our results experimentally. We demonstrate that, for real classification problems on MNIST and TIMIT datasets, our method combined with random sampling yields up to 3 times lower kernel approximation error. With sparse ANOVA kernels, our method slightly improves classification accuracy compared to the state-of-the-art kernel methods based on random Fourier features (which are already shown to match the performance of deep neural networks), all while speeding up the feature generation process.

## 2 Related Work

Much work has been done on extracting features for kernel methods. The random Fourier features method has been analyzed in the context of several learning algorithms, and its generalization error has been characterized and compared to that of other kernel-based algorithms [24]. It has also been compared to the Nyström method [35], which is data-dependent and thus can sometimes outperform random Fourier features. Other recent work has analyzed the generalization performance of the random Fourier features algorithm [17], and improved the bounds on its maximum error [29, 31].

While we focus here on deterministic approximations to the Fourier transform integral and compare them to Monte Carlo estimates, these are not the only two methods available to us. A possible middle-ground method is *quasi-Monte Carlo* estimation, in which low-discrepancy sequences, rather than the fully-random samples of Monte Carlo estimation, are used to approximate the integral. This approach was analyzed in Yang et al. [34] and shown to achieves an asymptotic error of $\epsilon = O\left(D^{-1}\left(\log(D)\right)^d\right)$. While this is asymptotically better than the random Fourier features method, the complexity of the quasi-Monte Carlo method coupled with its larger constant factors prevents it from being strictly better than its predecessor. Our method still requires asymptotically fewer samples as $\epsilon$ goes to 0.

Our deterministic approach here takes advantage of a long line of work on numerical quadrature for estimating integrals. Bach [1] analyzed in detail the connection between quadrature and random feature expansions, thus deriving bounds for the number of samples required to achieve a given average approximation error (though they did not present complexity results regarding maximum error nor suggested new feature maps). This connection allows us to leverage longstanding deterministic numerical integration methods such as Gaussian quadrature [6, 33] and sparse grids [2].

Unlike many other kernels used in machine learning, such as the Gaussian kernel, the sparse ANOVA kernel allows us to encode prior information about the relationships among the input variables into the kernel itself. Sparse ANOVA kernels have been shown [30] to work well for many classification tasks, especially in structural modeling problems that benefit from both the good generalization of a kernel machine and the representational advantage of a sparse model [9].

## 3    Kernels and Quadrature

We start with a brief overview of kernels. A kernel function $k \colon \mathbb{R}^d \times \mathbb{R}^d \to \mathbb{R}$ encodes the *similarity* between pairs of examples. In this paper, we focus on shift invariant kernels (those which satisfy $k(x,y) = k(x-y)$, where we overload the definition of $k$ to also refer to a function $k : \mathbb{R}^d \to \mathbb{R}$) that are positive definite and properly scaled. A kernel is positive definite if its Gram matrix is always positive definite for all non-trivial inputs, and it is properly-scaled if $k(x,x) = 1$ for all $x$. In this setting, our results make use of a theorem [25] that also provides the "key insight" behind the random Fourier features method.

**Theorem 1** (Bochner's theorem). *A continuous shift-invariant properly-scaled kernel $k : \mathbb{R}^d \times \mathbb{R}^d \to \mathbb{R}$ is positive definite if and only if $k$ is the Fourier transform of a proper probability distribution.*

We can then write $k$ in terms of its Fourier transform $\Lambda$ (which is a proper probability distribution):

$$k(x-y) = \int_{\mathbb{R}^d} \Lambda(\omega) \exp(j\omega^\top (x-y)) \, d\omega. \tag{1}$$

For $\omega$ distributed according to $\Lambda$, this is equivalent to writing

$$k(x-y) = \mathbf{E}\left[\exp(j\omega^\top(x-y))\right] = \mathbf{E}\left[\langle \exp(j\omega^\top x), \exp(j\omega^\top y)\rangle\right],$$

where we use the usual Hermitian inner product $\langle x,y\rangle = \sum_i x_i \overline{y_i}$. The random Fourier features method proceeds by estimating this expected value using Monte Carlo sampling averaged across $D$ random selections of $\omega$. Equivalently, we can think of this as approximating (1) with a discrete sum at randomly selected sample points.

Our objective is to choose some points $\omega_i$ and weights $a_i$ to uniformly approximate the integral (1) with $\tilde{k}(x-y) = \sum_{i=1}^{D} a_i \exp(j\omega_j^\top(x-y))$. To obtain a *feature map* $z : \mathbb{R}^d \to \mathbb{C}^D$ where $\tilde{k}(x-y) = \sum_{i=1}^{D} a_i z_i(x)\overline{z_i(y)}$, we can define

$$z(x) = \begin{bmatrix} \sqrt{a_1}\exp(j\omega_1^\top x) & \dots & \sqrt{a_D}\exp(j\omega_D^\top x) \end{bmatrix}^\top.$$

We aim to bound the maximum error for $x, y$ in a region $\mathcal{M}$ with diameter $M = \sup_{x,y\in\mathcal{M}} \|x-y\|$:

$$\epsilon = \sup_{(x,y)\in\mathcal{M}} \left|k(x-y) - \tilde{k}(x-y)\right| = \sup_{\|u\|\leq M} \left|\int_{\mathbb{R}^d} \Lambda(\omega)e^{j\omega^\top u}\, d\omega - \sum_{i=1}^{D} a_i e^{j\omega_i^\top u}\right|. \tag{2}$$

A *quadrature rule* is a choice of $\omega_i$ and $a_i$ to minimize this maximum error. To evaluate a quadrature rule, we are concerned with the *sample complexity* (for a fixed diameter $M$).

**Definition 1.** For any $\epsilon > 0$, a quadrature rule has sample complexity $D_{\mathrm{SC}}(\epsilon) = D$, where $D$ is the smallest value such that the rule, when instantiated with $D$ samples, has maximum error at most $\epsilon$.

We will now examine ways to construct deterministic quadrature rules and their sample complexities.

### 3.1    Gaussian Quadrature

Gaussian quadrature is one of the most popular techniques in one-dimensional numerical integration. The main idea is to approximate integrals of the form $\int \Lambda(\omega)f(\omega)\, d\omega \approx \sum_{i=1}^{D} a_i f(\omega_i)$ such that the approximation is exact for all polynomials below a certain degree; $D$ points are sufficient for polynomials of degree up to $2D-1$. While the points and weights used by Gaussian quadrature depend both on the distribution $\Lambda$ and the parameter $D$, they can be computed efficiently using orthogonal polynomials [10, 32]. Gaussian quadrature produces accurate results when integrating functions that are well-approximated by polynomials, which include all subgaussian densities.

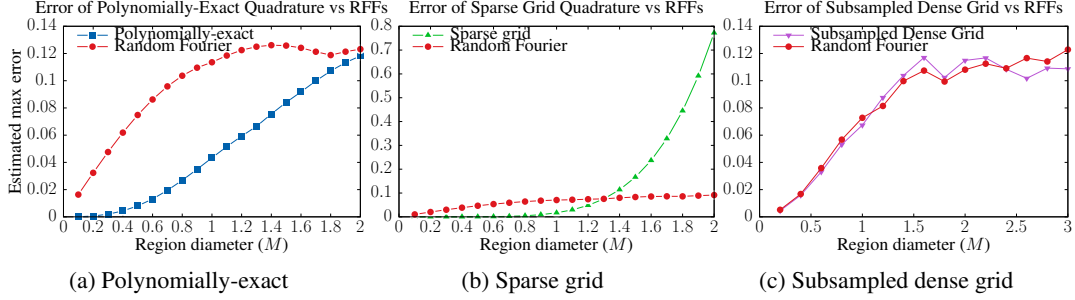

Figure 1: Error comparison (empirical maximum over $10^6$ uniformly-distributed samples) of different quadrature schemes and the random Fourier features method.

**Definition 2** (Subgaussian Distribution). We say that a distribution $\Lambda : \mathbb{R}^d \to \mathbb{R}$ is *subgaussian* with parameter $b$ if for $X \sim \Lambda$ and for all $t \in \mathbb{R}^d$, $\mathbf{E}\left[\exp(\langle t, X \rangle)\right] \leq \exp\left(\frac{1}{2}b^2 \|t\|^2\right)$.

We subsequently assume that the distribution $\Lambda$ is subgaussian, which is a technical restriction compared to random Fourier features. Many of the kernels encountered in practice have subgaussian spectra, including the ubiquitous Gaussian kernel. More importantly, we can approximate any kernel by convolving it with the Gaussian kernel, resulting in a subgaussian kernel. The approximation error can be made much smaller than the inherent noise in the data generation process.

## 3.2 Polynomially-Exact Rules

Since Gaussian quadrature is so successful in one dimension, as commonly done in the numerical analysis literature [14], we might consider using quadrature rules that are multidimensional analogues of Gaussian quadrature — rules that are accurate for all polynomials up to a certain degree $R$. In higher dimensions, this is equivalent to saying that our quadrature rule satisfies

$$\int_{\mathbb{R}^d} \Lambda(\omega) \prod_{l=1}^{d} (e_l^\top \omega)^{r_l} \, d\omega = \sum_{i=1}^{D} a_i \prod_{l=1}^{d} (e_l^\top \omega_i)^{r_l} \quad \text{for all } r \in \mathbb{N}^d \text{ such that } \sum_l r_l \leq R, \quad (3)$$

where $e_l$ are the standard basis vectors.

To test the accuracy of polynomially-exact quadrature, we constructed a feature map for a Gaussian kernel, $\Lambda(\omega) = (2\pi)^{-\frac{d}{2}} \exp\left(-\frac{1}{2} \|\omega\|^2\right)$, in $d = 25$ dimensions with $D = 1000$ and accurate for all polynomials up to degree $R = 2$. In Figure 1a, we compared this to a random Fourier features rule with the same number of samples, over a range of region diameters $M$ that captures most of the data points in practice (as the kernel is properly scaled). For small regions in particular, a polynomially-exact scheme can have a significantly lower error than a random Fourier feature map.

This experiment motivates us to investigate theoretical bounds on the behavior of this method. For subgaussian kernels, it is straightforward to bound the maximum error of a polynomially-exact feature map using the Taylor series approximation of the exponential function in (2).

**Theorem 2.** *Let $k$ be a kernel with $b$-subgaussian spectrum, and let $\tilde{k}$ be its estimation under some quadrature rule with non-negative weights that is exact up to some even degree $R$. Let $\mathcal{M} \subset \mathbb{R}^d$ be some region of diameter $M$. Then, for all $x, y \in \mathcal{M}$, the error of the quadrature features approximation is bounded by*

$$\left| k(x - y) - \tilde{k}(x - y) \right| \leq 3 \left( \frac{e b^2 M^2}{R} \right)^{\frac{R}{2}}.$$

All the proofs are found in the Appendix.

To bound the sample complexity of polynomially-exact quadrature, we need to determine how many quadrature samples we will need to satisfy the conditions of Theorem 2. There are $\binom{d+R}{d}$ constraints in (3), so a series of polynomially-exact quadrature rules that use only about this many sample points can yield a bound on the sample complexity of this quadrature rule.

**Corollary 1.** *Assume that we are given a class of feature maps that satisfy the conditions of Theorem 2, and that all have a number of samples $D \le \beta\binom{d+R}{d}$ for some fixed constant $\beta$. Then, for any $\gamma > 0$, the sample complexity of features maps in this class can be bounded by*

$$D(\epsilon) \le \beta 2^d \max\left(\exp\left(e^{2\gamma+1}b^2M^2\right), \left(\frac{3}{\epsilon}\right)^{\frac{1}{\gamma}}\right).$$

*In particular, for a fixed dimension d, this means that for any $\gamma$, $D(\epsilon) = O\left(\epsilon^{-\frac{1}{\gamma}}\right)$.*

The result of this corollary implies that, in terms of the desired error $\epsilon$, the sample complexity increases asymptotically slower than any negative power of $\epsilon$. Compared to the result for random Fourier features which had $D(\epsilon) = O(\epsilon^{-2})$, this has a much weaker dependence on $\epsilon$. While this weaker dependence does come at the cost of an additional factor of $2^d$, it is a constant cost of operating in dimension $d$, and is not dependent on the error $\epsilon$.

The more pressing issue, when comparing polynomially-exact features to random Fourier features, is the fact that we have no way of efficiently constructing quadrature rules that satisfy the conditions of Theorem 2. One possible construction involves selecting random sample points $\omega_i$, and then solving (3) for the values of $a_i$ using a non-negative least squares (NNLS) algorithm. While this construction works in low dimensions — it is the method we used for the experiment in Figure 1a — it rapidly becomes infeasible to solve for higher values of $d$ and $R$.

We will now show how to overcome this issue by introducing quadrature rules that can be rapidly constructed using grid-based quadrature rules. These rules are constructed directly from products of a one-dimensional quadrature rule, such as Gaussian quadrature, and so avoid the construction-difficulty problems encountered in this section. Although grid-based quadrature rules can be constructed for any kernel function [2], they are easier to conceptualize when the kernel $k$ factors along the dimensions, as $k(u) = \prod_{i=1}^d k_i(u_i)$. For simplicity we will focus on this factorizable case.

### 3.3 Dense Grid Quadrature

The simplest way to do this is with a *dense grid* (also known as tensor product) construction. A dense grid construction starts by factoring the integral (1) into $k(u) = \prod_{i=1}^d \left(\int_{-\infty}^{\infty} \Lambda_i(\omega)\exp(j\omega e_i^\top u)\,d\omega\right)$, where $e_i$ are the standard basis vectors. Since each of the factors is an integral over a single dimension, we can approximate them all with a one-dimensional quadrature rule. In this paper, we focus on Gaussian quadrature, although we could also use other methods such as Clenshaw-Curtis [3]. Taking tensor products of the points and weights results in the dense grid quadrature. The detailed construction is given in Appendix A.

The individual Gaussian quadrature rules are exact for all polynomials up to degree $2L - 1$, so the dense grid is also accurate for all such polynomials. Theorem 2 then yields a bound on its sample complexity.

**Corollary 2.** *Let $k$ be a kernel with a spectrum that is subgaussian with parameter $b$. Then, for any $\gamma > 0$, the sample complexity of dense grid features can be bounded by*

$$D(\epsilon) \le \max\left(\exp\left(de^{\gamma d}\frac{eb^2M^2}{2}\right), \left(\frac{3}{\epsilon}\right)^{\frac{1}{\gamma}}\right).$$

*In particular, as was the case with polynomially-exact features, for a fixed d, $D(\epsilon) = O\left(\epsilon^{-\frac{1}{\gamma}}\right)$.*

Unfortunately, this scheme suffers heavily from the curse of dimensionality, since the sample complexity is doubly-exponential in $d$. This means that, even though they are easy to compute, the dense grid method does not represent a useful solution to the issue posed in Section 3.2.

### 3.4 Sparse Grid Quadrature

The curse of dimensionality for quadrature in high dimensions has been studied in the numerical integration setting for decades. One of the more popular existing techniques for getting around

the curse is called *sparse grid* or Smolyak quadrature [28], originally developed to solve partial differential equations. Instead of taking the tensor product of the one-dimensional quadrature rule, we only include points up to some fixed total level $A$, thus constructing a linear combination of dense grid quadrature rules that achieves a similar error with exponentially fewer points than a single larger quadrature rule. The detailed construction is given in Appendix B. Compared to polynomially-exact rules, sparse grid quadrature can be computed quickly and easily (see Algorithm 4.1 from Holtz [12]).

To measure the performance of sparse grid quadrature, we constructed a feature map for the same Gaussian kernel analyzed in the previous section, with $d = 25$ dimensions and up to level $A = 2$. We compared this to a random Fourier features rule with the same number of samples, $D = 1351$, and plot the results in Figure 1b. As was the case with polynomially-exact quadrature, this sparse grid scheme has tiny error for small-diameter regions, but this error unfortunately increases to be even larger than that of random Fourier features as the region diameter increases.

The sparse grid construction yields a bound on the sample count: $D \leq 3^A \binom{d+A}{A}$, where $A$ is the bound on the total level. By extending known bounds on the error of Gaussian quadrature, we can similarly bound the error of the sparse grid feature method.

**Theorem 3.** *Let $k$ be a kernel with a spectrum that is subgaussian with parameter $b$, and let $\tilde{k}$ be its estimation under the sparse grid quadrature rule up to level $A$. Let $\mathcal{M} \subset \mathbb{R}^d$ be some region of diameter $M$, and assume that $A \geq 24eb^2M^2$. Then, for all $x, y \in \mathcal{M}$, the error of the quadrature features approximation is bounded by*

$$\left| k(x-y) - \tilde{k}(x-y) \right| \leq 2^d \left( \frac{12eb^2M^2}{A} \right)^A.$$

This, along with our above upper bound on the sample count, yields a bound on the sample complexity.

**Corollary 3.** *Let $k$ be a kernel with a spectrum that is subgaussian with parameter $b$. Then, for any $\gamma > 0$, the sample complexity of sparse grid features can be bounded by*

$$D(\epsilon) \leq 2^d \max \left( \exp \left( 24e^{2\gamma+1}b^2M^2 \right), 2^{\frac{d}{\gamma}} \epsilon^{-\frac{1}{\gamma}} \right).$$

*As was the case with all our previous deterministic features maps, for a fixed $d$, $D(\epsilon) = O \left( \epsilon^{-\frac{1}{\gamma}} \right)$.*

**Subsampled grids**    One of the downsides of the dense/sparse grids analyzed above is the difficulty of tuning the number of samples extracted in the feature map. As the only parameter we can typically set is the degree of polynomial exactness, even a small change in this (e.g., from 2 to 4) can produce a significant increase in the number of features. However, we can always subsample the grid points according to the distribution determined by their weights to both tame the curse of dimensionality and to have fine-grained control over the number of samples. For simplicity, we focus on subsampling the dense grid. In Figure 1c, we compare the empirical errors of subsampled dense grid and random Fourier features, noting that they are essentially the same across all diameters.

### 3.5   Reweighted Grid Quadrature

Both random Fourier features and dense/sparse grid quadratures are data-independent. We now describe a data-adaptive method to choose a quadrature for a pre-specified number of samples: reweighting the grid points to minimize the difference between the approximate and the exact kernel on a small subset of data. Adjusting the grid to the data distribution yields better kernel approximation.

We approximate the kernel $k(x - y)$ with

$$\tilde{k}(x-y) = \sum_{i=1}^{D} a_i \exp(j\omega_i^\top (x-y)) = \sum_{i=1}^{D} a_i \cos(\omega_i^\top (x-y)),$$

where $a_i \geq 0$, as $k$ is real-valued. We first choose the set of potential grid points $\omega_1, \ldots, \omega_D$ by sampling from a dense grid of Gaussian quadrature points. To solve for the weights $a_1, \ldots, a_D$, we independently sample $n$ pairs $(x_1, y_1), \ldots, (x_n, y_n)$ from the dataset, then minimize the empirical mean squared error (with variable $a_1, \ldots, a_D$):

$$\begin{aligned} \text{minimize} \quad & \frac{1}{n} \sum_{l=1}^{n} \left( k(x_l - y_l) - \tilde{k}(x_l - y_l) \right)^2 \\ \text{subject to} \quad & a_i \geq 0, \text{ for } i = 1, \ldots, D. \end{aligned}$$

For appropriately defined matrix $M$ and vector $b$, this is an NNLS problem of minimizing $\frac{1}{n} \|Ma - b\|^2$ subject to $a \geq 0$, with variable $a \in \mathbb{R}^D$. The solution is often sparse, due to the active elementwise constraints $a \geq 0$. Hence we can pick a larger set of potential grid points $\omega_1, \ldots, \omega_{D'}$ (with $D' > D$) and solve the above problem to obtain a smaller set of grid points (those with $a_j > 0$). To get even sparser solution, we add an $\ell_1$-penalty term with parameter $\lambda \geq 0$:

$$\begin{aligned} \text{minimize} \quad & \tfrac{1}{n} \|Ma - b\|^2 + \lambda \mathbf{1}^\top a \\ \text{subject to} \quad & a_i \geq 0, \text{ for } i = 1, \ldots, D'. \end{aligned}$$

Bisecting on $\lambda$ yields the desired number of grid points.

As this is a data-dependent quadrature, we empirically evaluate its performance on the TIMIT dataset, which we will describe in more details in Section 5. In Figure 2b, we compare the estimated root mean squared error on the dev set of different feature generation schemes against the number of features $D$ (mean and standard deviation over 10 runs). Random Fourier features, Quasi-Monte Carlo (QMC) with Halton sequence, and subsampled dense grid have very similar approximation error, while reweighted quadrature has much lower approximation error. Reweighted quadrature achieves 2–3 times lower error for the same number of features and requiring 3–5 times fewer features for a fixed threshold of approximation error compared to random Fourier features. Moreover, reweighted features have extremely low variance, even though the weights are adjusted based only on a very small fraction of the dataset (500 samples out of 1 million data points).

**Faster feature generation**   Not only does grid-based quadrature yield better statistical performance to random Fourier features, it also has some notable systems benefits. Generating quadrature features requires a much smaller number of multiplies, as the grid points only take on a finite set of values for all dimensions (assuming an isotropic kernel). For example, a Gaussian quadrature that is exact up to polynomials of degree 21 only requires 11 grid points for each dimension. To generate the features, we multiply the input with these 11 numbers before adding the results to form the deterministic features. The save in multiples may be particularly significant in architectures such as application-specific integrated circuits (ASICs). In our experiment on the TIMIT dataset in Section 5, this specialized matrix multiplication procedure (on CPU) reduces the feature generation time in half.

## 4   Sparse ANOVA Kernels

One type of kernel that is commonly used in machine learning, for example in structural modeling, is the *sparse ANOVA kernels* [11, 8]. They are also called *convolutional kernels*, as they operate similarly to the convolutional layer in CNNs. These kernels have achieved state-of-the-art performance on large real-world datasets [18, 22], as we will see in Section 5. A kernel of this type can be written as

$$k(x, y) = \sum_{S \in \mathcal{S}} \prod_{i \in S} k_1(x_i - y_i),$$

where $\mathcal{S}$ is a set of subsets of the variables in $\{1, \ldots, d\}$, and $k_1$ is a one-dimensional kernel. (Straightforward extensions, which we will not discuss here, include using different one-dimensional kernels for each element of the products, and weighting the sum.)  Sparse ANOVA kernels are used to encode sparse dependencies among the variables: two variables are related if they appear together in some $S \in \mathcal{S}$. These sparse dependencies are typically problem-specific: each $S$ could correspond to a factor in the graph if we are analyzing a distribution modeled with a factor graph. Equivalently, we can think of the set $\mathcal{S}$ as a hypergraph, where each $S \in \mathcal{S}$ corresponds to a hyperedge. Using this notion, we define the *rank* of an ANOVA kernel to be $r = \max_{S \in \mathcal{S}} |S|$, the *degree* as $\Delta = \max_{i \in \{1, \ldots, d\}} |\{S \in \mathcal{S} | i \in S\}|$, and the *size* of the kernel to be the number of hyperedges $m = |\mathcal{S}|$. For sparse models, it is common for both the rank and the degree to be small, even as the number of dimensions $d$ becomes large, so $m = O(d)$. This is the case we focus on in this section.

It is straightforward to apply the random Fourier features method to construct feature maps for ANOVA kernels: construct feature maps for each of the (at most $r$-dimensional) sub-kernels $k_S(x - y) = \prod_{i \in S} k_1(x_i - y_i)$ individually, and then combine the results. To achieve overall error $\epsilon$, it suffices for each of the sub-kernel feature maps to have error $\epsilon/m$; this can be achieved by random Fourier features using $D_S = \tilde{\Omega}\left(r(\epsilon m^{-1})^{-2}\right) = \tilde{\Omega}\left(rm^2\epsilon^{-2}\right)$ samples each, where the notation $\tilde{\Omega}$ hides the $\log 1/\epsilon$ factor.  Summed across all the $m$ sub-kernels, this means that the random

Fourier features map can achieve error $\epsilon$ with constant probability with a sample complexity of $D(\epsilon) = \tilde{\Omega}\left(rm^3\epsilon^{-2}\right)$ samples. While it is nice to be able to tackle this problem using random features, the cubic dependence on $m$ in this expression is undesirable: it is significantly larger than the $D = \tilde{\Omega}(d\epsilon^{-2})$ we get in the non-ANOVA case.

Can we construct a deterministic feature map that has a better error bound? It turns out that we can.

**Theorem 4.** *Assume that we use polynomially-exact quadrature to construct features for each of the sub-kernels $k_S$, under the conditions of Theorem 2, and then combine the resulting feature maps to produce a feature map for the full ANOVA kernel. For any $\gamma > 0$, the sample complexity of this method is*

$$D(\epsilon) \leq \beta m 2^r \max\left(\exp\left(e^{2\gamma+1}b^2M^2\right), (3\Delta)^{\frac{1}{\gamma}}\epsilon^{-\frac{1}{\gamma}}\right).$$

Compared to the random Fourier features, this rate depends only linearly on $m$. For fixed parameters $\beta$, $b$, $M$, $\Delta$, $r$, and for any $\gamma > 0$, we can bound the sample complexity $D(\epsilon) = O(m\epsilon^{-\frac{1}{\gamma}})$, which is better than random Fourier features *both* in terms of the kernel size $m$ and the desired error $\epsilon$.

## 5 Experiments

To evaluate the performance of deterministic feature maps, we analyzed the accuracy of a sparse ANOVA kernel on the MNIST digit classification task [16] and the TIMIT speech recognition task [5].

**Digit classification on MNIST**   This task consists of $70,000$ examples ($60,000$ in the training dataset and $10,000$ in the test dataset) of hand-written digits which need to be classified. Each example is a $28 \times 28$ gray-scale image. Clever kernel-based SVM techniques are known to achieve very low error rates (e.g., $0.79\%$) on this problem [20]. We do not attempt to compare ourselves with these rates; rather, we compare random Fourier features and subsampled dense grid features that both approximate the same ANOVA kernel. The ANOVA kernel we construct is designed to have a similar structure to the first layer of a convolutional neural network [27]. Just as a filter is run on each $5 \times 5$ square of the image, for our ANOVA kernel, each of the sub-kernels is chosen to run on a $5 \times 5$ square of the original image (note that there are many, $(28 - 5 + 1)^2 = 576$, such squares). We choose the simple Gaussian kernel as our one-dimensional kernel.

Figure 2a compares the dense grid subsampling method to random Fourier features across a range of feature counts. The deterministic feature map with subsampling performs better than the random Fourier feature map across most large feature counts, although its performance degrades for very small feature counts. The deterministic feature map is also somewhat faster to compute, taking—for the 28800-features—320 seconds vs. 384 seconds for the random Fourier features, a savings of $17\%$.

**Speech recognition on TIMIT**   This task requires producing accurate transcripts from raw audio recordings of conversations in English, involving 630 speakers, for a total of 5.4 hours of speech. We use the kernel features in the acoustic modeling step of speech recognition. Each data point corresponds to a frame (10ms) of audio data, preprocessed using the standard feature space Maximum Likelihood Linear Regression (fMMLR) [4]. The input $x$ has dimension 40. After generating kernel features $z(x)$ from this input, we model the corresponding phonemes $y$ by a multinomial logistic regression model. Again, we use a sparse ANOVA kernel, which is a sum of 50 sub-kernels of the form $\exp(-\gamma\|x_S - y_S\|^2)$, each acting on a subset $S$ of 5 indices. These subsets are randomly chosen a priori. To reweight the quadrature features, we sample 500 data points out of 1 million.

We plot the phone error rates (PER) of a speech recognizer trained based on different feature generation schemes against the number of features $D$ in Figure 2c (mean and standard deviation over 10 runs). Again, subsampled dense grid performs similarly to random Fourier features, QMC yields slightly higher error, while reweighted features achieve slightly lower phone error rates. All four methods have relatively high variability in their phone error rates due to the stochastic nature of the training and decoding steps in the speech recognition pipeline. The quadrature-based features (subsampled dense grids and reweighted quadrature) are about twice as fast to generate, compared to random Fourier features, due to the small number of multiplies required. We use the same setup as May et al. [22], and the performance here matches both that of random Fourier features and deep neural networks in May et al. [22].

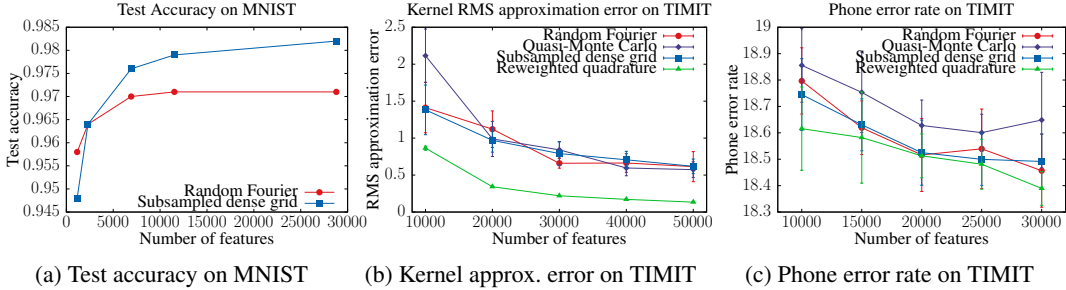

(a) Test accuracy on MNIST    (b) Kernel approx. error on TIMIT    (c) Phone error rate on TIMIT

Figure 2: Performance of different feature generation schemes on MNIST and TIMIT.

# 6 Conclusion

We presented deterministic feature maps for kernel machines. We showed that we can achieve better scaling in the desired accuracy $\epsilon$ compared to the state-of-the-art method, random Fourier features. We described several ways to construct these feature maps, including polynomially-exact quadrature, dense grid construction, sparse grid construction, and reweighted grid construction. Our results apply well to the case of sparse ANOVA kernels, achieving significant improvements (in the dependency on the dimension $d$) over random Fourier features. Finally, we evaluated our results experimentally, and showed that ANOVA kernels with deterministic feature maps can produce comparable accuracy to the state-of-the-art methods based on random Fourier features on real datasets.

ANOVA kernels are an example of how structure can be used to define better kernels. Resembling the convolutional layers of convolutional neural networks, they induce the necessary inductive bias in the learning process. Given CNNs' recent success in other domains beside images, such as sentence classification [15] and machine translation [7], we hope that our work on deterministic feature maps will enable kernel methods such as ANOVA kernels to find new areas of application.

### Acknowledgments

This material is based on research sponsored by Defense Advanced Research Projects Agency (DARPA) under agreement number FA8750-17-2-0095. We gratefully acknowledge the support of the DARPA SIMPLEX program under No. N66001-15-C-4043, DARPA FA8750-12-2-0335 and FA8750-13-2-0039, DOE 108845, National Institute of Health (NIH) U54EB020405, the National Science Foundation (NSF) under award No. CCF-1563078, the Office of Naval Research (ONR) under awards No. N000141210041 and No. N000141310129, the Moore Foundation, the Okawa Research Grant, American Family Insurance, Accenture, Toshiba, and Intel. This research was supported in part by affiliate members and other supporters of the Stanford DAWN project: Intel, Microsoft, Teradata, and VMware. The U.S. Government is authorized to reproduce and distribute reprints for Governmental purposes notwithstanding any copyright notation thereon. The views and conclusions contained herein are those of the authors and should not be interpreted as necessarily representing the official policies or endorsements, either expressed or implied, of DARPA or the U.S. Government. Any opinions, findings, and conclusions or recommendations expressed in this material are those of the authors and do not necessarily reflect the views of DARPA, AFRL, NSF, NIH, ONR, or the U.S. government.

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
