[Supplementary Material]

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

## A   Dense grid construction

If we let $\int_{-\infty}^{\infty} \Lambda_i(\omega) f(\omega)\, d\omega \approx \sum_{l=1}^{L_i} a_{i,l} f(\omega_{i,l})$ be the Gaussian quadrature rule for each integral, then we can approximate $k$ with

$$\tilde{k}(u) = \prod_{i=1}^{d} \sum_{l=1}^{L_k} a_{i,l} \exp(j\omega_{i,l} e_i^\top u).$$

If we define $a_{\mathbf{l}} = \prod_{i=1}^{d} a_{i,l_i}$ and $\omega_{\mathbf{l}} = \sum_{i=1}^{d} \omega_{i,l_i} e_i$ then we are left with the tensor product quadrature rule

$$\tilde{k}(u) = \sum_{\mathbf{l} \in \prod_{i=1}^{d}\{1 \ldots L_i\}} a_{\mathbf{l}} \exp\left(j\omega_{\mathbf{l}}^\top u\right) \tag{4}$$

over $D = \prod L_i$ points — we can simplify this to $L^d$ in the case where every $L_i = L$.

## B   Sparse grid construction

Here, we briefly describe the sparse grid construction. We start by letting let $G_i^L(u_i)$ be the approximation of $k_i(u_i)$ that results from applying the one-dimensional Gaussian quadrature rule with $L$ points: for the appropriate sample points and weights,

$$G_i^L(u_i) = \sum_{l=1}^{L} a_l \exp(j u_i \omega_l).$$

One of the properties of Gaussian quadrature is that it is exact in the limit of large $L$. In particular, this limit means that we can decompose $k_i(u_i)$ as the infinite sum

$$k_i(u_i) = G_i^1(u_i) + \sum_{m=1}^{\infty} \left(G_i^{2^m}(u_i) - G_i^{2^{m-1}}(u_i)\right) = \sum_{m=0}^{\infty} \Delta_{i,m}(u_i),$$

where $\Delta_{i,m}(u_i) = G_i^{2^m}(u_i) - G_i^{2^{m-1}}(u_i)$. To represent $k(u)$, it suffices to use the product

$$k(u) = \sum_{\mathbf{m} \in \mathbb{N}^d} \prod_{i=1}^{d} \Delta_{i,m_i}(u_i) = \sum_{\mathbf{m} \in \mathbb{N}^d} \Delta_{\mathbf{m}}(u)$$

where $\Delta_{\mathbf{m}}(u) = \prod_{i=1}^{d} \Delta_{i,m_i}(u_i)$. We can think of these $\Delta_{\mathbf{m}}$ forming a "grid" of terms in $\mathbb{N}^d$. We plot this grid for $d = 2$ in Figure 3. The dense grid approximation is equivalent to summing up a hypercube of these terms, which we illustrate as a square in the figure.

Figure 3: Grid of approximation terms $\Delta_{\mathbf{m}}$.

Smolyak's sparse grid approximation approximates this sum by using only those $\Delta_{\mathbf{m}}$ that can be computed with a "small" number of samples. Specifically, the sparse grid up to level $A$ is defined as,

$$\tilde{k}(u) = \sum_{\mathbf{m} \in \mathbb{N}^d, \, \mathbf{1}^\top \mathbf{m} \le A} \Delta_{\mathbf{m}}(u).$$

In Figure 3, this is illustrated by the blue triangle — the efficiency of sparse grids comes from the fact that in higher dimensions, the simplex of terms used by the sparse grid contains exponentially (in $d$) fewer quadrature points than the hypercube of terms used by a dense grid.

Now, for any $u$, each $\Delta_{\mathbf{m}}(u)$ can be computed using the tensor product quadrature rule from (4); the number of samples required is no greater than $3^{\mathbf{1}^\top \mathbf{m}}$. Combining this with the previous equation gives us a rough upper bound on the sample count of the sparse grid construction

$$D \le \sum_{\mathbf{m} \in \mathbb{N}^d, \, \mathbf{1}^\top \mathbf{m} \le A} 3^{\mathbf{1}^\top \mathbf{m}} \le 3^A \binom{d + A}{A}.$$

## C  Proofs

### C.1  Proof of Theorem 2

In order to prove this theorem, we will need a couple of lemmas.

**Lemma 1** (Stirling's Approximation). *For any positive integer $n$,*

$$\left(n + \frac{1}{2}\right) \log n - n + \frac{1}{2} \log(2\pi) + \frac{1}{12n + 1} \le \log n! \le \left(n + \frac{1}{2}\right) \log n - n + \frac{1}{2} \log(2\pi) + \frac{1}{12n}.$$

**Lemma 2** (Subgaussian Moment Bound). *If a random variable $X$ is $b$-subgaussian, then its $p$-th moment is bounded by*

$$\mathbf{E}\left[\|X\|^p\right] \le p 2^{\frac{p}{2}} b^p \Gamma\left(\frac{p}{2}\right).$$

We now prove the theorem.

*Proof of Theorem 2.* For any $x$, define $\epsilon(x)$, the error function, as

$$\epsilon(x) = \left| k(x) - \sum_{i=1}^{D} a_i \exp(jx^\top \omega_i) \right|.$$

By Taylor's theorem, there exists a function $\beta(z)$ such that

$$\exp(jz) = \sum_{k=0}^{R-1} \frac{(jz)^k}{k!} + \frac{(jz)^R}{R!} \exp(j\beta(z)).$$

This is the mean value theorem form for the Taylor series remainder. Therefore we can write $\epsilon(x)$ as

$$\left| k(x) - \sum_{i=1}^{D} a_i \exp(jx^\top \omega_i) \right|$$

$$= \left| \int \Lambda(\omega) \exp(jx^\top \omega) d\omega - \sum_{i=1}^{D} a_i \exp(jx^\top \omega_i) \right|$$

$$= \left| \int \Lambda(\omega) \left( \sum_{l=0}^{R-1} \frac{(jx^\top \omega)^l}{l!} + \frac{(jx^\top \omega)^R}{R!} e^{j\beta(x^\top \omega)} \right) d\omega - \sum_{i=1}^{D} a_i \left( \sum_{l=0}^{R-1} \frac{(jx^\top \omega_i)^l}{l!} + \frac{(jx^\top \omega_i)^R}{R!} e^{j\beta(x^\top \omega_i)} \right) \right|$$

$$= \left| \sum_{l=0}^{R-1} \frac{j^l}{l!} \left( \int \Lambda(\omega)(x^\top \omega)^l d\omega - \sum_{i=1}^{D} a_i (x^\top \omega_i)^l \right) + \frac{j^R}{R!} \left( \int \Lambda(\omega)(x^\top \omega)^R e^{j\beta(x^\top \omega)} d\omega - \sum_{i=1}^{D} a_i (x^\top \omega_i)^R e^{j\beta(x^\top \omega_i)} \right) \right|.$$

Now, since our quadrature is exact up to degree $R$, by the condition from (3), the first term is zero:

$$\epsilon(x) = \left| \frac{j^R}{R!} \left( \int \Lambda(\omega)(x^\top \omega)^R \exp(j\beta(x^\top \omega)) d\omega - \sum_{i=1}^{D} a_i (x^\top \omega_i)^R \exp(j\beta(x^\top \omega_i)) \right) \right|$$

$$\leq \frac{1}{R!} \left( \int \left| \Lambda(\omega)(x^\top \omega)^R \exp(j\beta(x^\top \omega)) \right| d\omega + \sum_{i=1}^{D} \left| a_i (x^\top \omega_i)^R \exp(j\beta(x^\top \omega_i)) \right| \right)$$

$$\leq \frac{1}{R!} \left( \int \left| \Lambda(\omega)(x^\top \omega)^R \right| d\omega + \sum_{i=1}^{D} \left| a_i (x^\top \omega_i)^R \right| \right).$$

Since $R$ is even, $a_i \geq 0$, and $\Lambda(\omega) \geq 0$,

$$\epsilon(x) \leq \frac{1}{R!} \left( \int \Lambda(\omega)(x^\top \omega)^R d\omega + \sum_{i=1}^{D} a_i (x^\top \omega_i)^R \right).$$

Again applying our condition from (3),

$$\epsilon(x) \leq \frac{2}{R!} \int \Lambda(\omega)(x^\top \omega)^R d\omega.$$

Finally, by Cauchy-Schwarz,

$$\epsilon(x) \leq \frac{2 \|x\|^R}{R!} \int \Lambda(\omega) \|\omega\|^R d\omega \leq \frac{2 \|x\|^R}{R!} \mathbf{E}_\Lambda \left[ \|\omega\|^R \right].$$

Now, since we assumed that $\Lambda$ was $b$-subgaussian, we can apply Lemma 2 to bound this expected value with

$$\epsilon(x) \leq \frac{2 \|x\|^R}{R!} \int \Lambda(\omega) \|\omega\|^R d\omega \leq \frac{2 \|x\|^R}{R!} R 2^{\frac{R}{2}} b^R \Gamma \left( \frac{R}{2} \right) = 4 b^R \|x\|^R \frac{2^{R/2}(R/2)!}{R!}.$$

Now we need to bound $\frac{2^{R/2}(R/2)!}{R!}$ using Stirling's approximation (Lemma 1):

$$-\log(R!) + \frac{R}{2}\log 2 + \log\left((R/2)!\right)$$

$$\leq - \left( \left(R + \frac{1}{2}\right)\log R - R + \frac{1}{2}\log(2\pi) + \frac{1}{12R+1} \right) + \frac{R}{2}\log 2 + \left(\frac{R}{2} + \frac{1}{2}\right)\log\left(\frac{R}{2}\right) - \frac{R}{2} + \frac{1}{2}\log(2\pi) + \frac{1}{6R}$$

$$= -R\log R - \frac{1}{2}\log R + \frac{R}{2} - \frac{1}{12R+1} + \frac{R}{2}\log 2 + \frac{R}{2}\log R + \frac{1}{2}\log R - \frac{R}{2}\log 2 - \frac{1}{2}\log 2 + \frac{1}{6R}$$

$$= \frac{R}{2} - \frac{1}{2}\log 2 - \frac{R}{2}\log R + \frac{1}{6R} - \frac{1}{12R+1}$$

$$\leq -\frac{1}{2}\log 2 + \frac{1}{12} - \frac{1}{25} + \frac{R}{2}\left(1 - \log R\right),$$

where we have used the fact that $R \geq 2$ since $R$ is even. Taking the exponential results in

$$\epsilon(x) \leq 4 b^R \|x\|^R \frac{e^{1/12 - 1/25}}{\sqrt{2}} \left(\frac{e}{R}\right)^{R/2} \leq 3 \left(\frac{eb^2 \|x\|^2}{R}\right)^{\frac{R}{2}}.$$

Therefore, for any $x, y \in \mathcal{M}$,

$$\left| k(x-y) - \tilde{k}(x-y) \right| = \left| k(x-y) - \sum_{i=1}^{D} a_i \exp(j(x-y)^\top \omega_i) \right|$$

$$= \epsilon(x-y)$$

$$\leq 3 \left(\frac{eb^2 \|x-y\|^2}{R}\right)^{\frac{R}{2}}.$$

Finally, since $\mathcal{M}$ has diameter $M$, we know that $\|x - y\| \le M$, so we can conclude that

$$\left| k(x, y) - \tilde{k}(x - y) \right| \le 3 \left( \frac{eb^2 M^2}{R} \right)^{\frac{R}{2}},$$

which is the desired expression.

$\square$

Using this, we can directly prove Corollary 1.

*Proof of Corollary 1.* By assumption, the number of samples required is

$$D \le \beta \binom{d + R}{d} \le \beta 2^d \cdot 2^R \le \beta 2^d \exp(R).$$

In order to ensure

$$\sup_{\|u\| \le M} \left| k(u) - \tilde{k}(u) \right| \le \epsilon,$$

it suffices by the result of Theorem 2 to have $R$ large enough that

$$3 \left( \frac{eb^2 M^2}{R} \right)^{R/2} \le \epsilon \qquad \text{and} \qquad \frac{eb^2 M^2}{R} < 1.$$

Suppose that we set $R$ such that

$$\frac{eb^2 M^2}{R} \le \exp(-2\gamma).$$

If $\gamma > 0$, then the second condition will be trivially satisfied. The first condition will also be satisfied when we set $R$ large enough that

$$3 \exp(-\gamma R) \le \epsilon.$$

This occurs when

$$\exp(R) \ge \left( \frac{3}{\epsilon} \right)^{1/\gamma}.$$

For this condition to hold, as $D \le \beta 2^d \exp(R)$, it suffices to have

$$D \ge \beta 2^d \left( \frac{3}{\epsilon} \right)^{1/\gamma}$$

samples. On the other hand, for $R$ to satisfy our original condition, we also need

$$R \ge eb^2 M^2 \exp(2\gamma).$$

This can be achieved when

$$D \ge \beta 2^d \exp(e^{2\gamma + 1} b^2 M^2).$$

Combining these two conditions using a maximum proves the corollary. $\square$

We can similarly prove Corollary 2.

*Proof of Corollary 2.* For the quadrature rule to be exact for polynomials of degrees up to (even) $R$, it suffices for each one-dimensional rule to have $L = R/2 + 1$ points. The total number of points used is then $D = L^d = \exp(d \log(R/2 + 1))$. Since $R \ge 2$ as it is even, $\log(R/2 + 1) \le R/2$, so $D \le \exp(Rd/2)$.

As in the proof of Corollary 1, to ensure $\sup_{\|u\| \le M} \left| k(u) - \tilde{k}(u) \right| \le \epsilon$, it suffices by the result of Theorem 2 to have $R$ large enough that

$$3 \left( \frac{eb^2 M^2}{R} \right)^{R/2} \le \epsilon \qquad \text{and} \qquad \frac{eb^2 M^2}{R} < 1.$$

Suppose that we set $R$ such that

$$\frac{eb^2 M^2}{R} \le \exp(-d\gamma).$$

If $\gamma > 0$, then the second condition will be trivially satisfied. The first condition will also be satisfied when we set $R$ large enough that

$$3 \exp(-\gamma R d/2) \le \epsilon.$$

This occurs when

$$\exp(Rd/2) \ge \left(\frac{3}{\epsilon}\right)^{1/\gamma}.$$

For this condition to hold, as $D \le \exp(Rd/2)$, it suffices to have

$$D \ge \left(\frac{3}{\epsilon}\right)^{1/\gamma}$$

samples. On the other hand, for $R$ to satisfy our original condition, we also need

$$R \ge eb^2 M^2 \exp(d\gamma).$$

This can be achieved when

$$D \ge \exp(de^{d\gamma} eb^2 M^2/2).$$

Combining these two conditions using a maximum proves the corollary. □

## C.2    Proof of Theorem 3

*Proof of Theorem 3.* Based on the construction of the sparse grid in Section B, $k$ and $\tilde{k}$ differ in the terms $\sum_{\mathbf{m} \in \mathbb{N}^d, \mathbf{1}^\top \mathbf{m} > A} \Delta_{\mathbf{m}}(u)$. Thus we need to bound the error

$$\sup_{\|u\| \le M} \left| k(u) - \tilde{k}(u) \right| = \sup_{\|u\| \le M} \left| \sum_{\mathbf{m} \in \mathbb{N}^d, \mathbf{1}^\top \mathbf{m} > A} \Delta_{\mathbf{m}}(u) \right| \le \sum_{\mathbf{m} \in \mathbb{N}^d, \mathbf{1}^\top \mathbf{m} > A} \sup_{\|u\| \le M} |\Delta_{\mathbf{m}}(u)|.$$

But $\Delta_{\mathbf{m}}(u)$ is just a product of one-dimensional rules, and we can apply Theorem 2 for each dimension. Indeed, the Gaussian quadrature rule with $L$ points $G_i^L$ is exact for polynomials of degree up to $2L - 1$, so the bound from Theorem 2 with $R = 2(L-1)$ becomes $3 \left(\frac{eb^2 u_i^2}{2(L-1)}\right)^{L-1} \le$ $3 \left(\frac{eb^2 u_i^2}{L}\right)^{L-1}$ (since $2(L-1) \ge L$). As $\Delta_{i,m_i}(u_i) = G_i^{2^{m_i}}(u_i) - G_i^{2^{m_i-1}}(u_i)$, we have

$$\begin{aligned}
|\Delta_{i,m_i}(u_i)| &= \left| G_i^{2^{m_i}}(u_i) - G_i^{2^{m_i-1}}(u_i) \right| \\
&\le \left| G_i^{2^{m_i}}(u_i) - k_i(u_i) \right| + \left| k_i(u_i) - G_i^{2^{m_i-1}}(u_i) \right| \\
&\le 3 \left(eb^2 u_i^2\right)^{2^{m_i}-1} 2^{-m_i(2^{m_i}-1)} + 3 \left(eb^2 u_i^2\right)^{2^{m_i-1}-1} 2^{-(m_i-1)(2^{m_i-1}-1)} \\
&\le 6 \left(eb^2 u_i^2\right)^{2^{m_i-1}} 2^{-(m_i-1)(2^{m_i-1}-1)} \\
&= 6 (\sqrt{e} b u_i)^{2^{m_i}} 2^{-(m_i-1)2^{m_i-1}} 2^{m_i-1}.
\end{aligned}$$

If we let $c_i = 2^{m_i-1}$ (and $c_i = 0$ if $m_i = 0$), then we can rewrite this as

$$|\Delta_{i,m_i}(u_i)| \le \frac{6(\sqrt{e} b)^{2c_i} c_i}{c_i^{c_i}} u_i^{2c_i}.$$

Thus

$$|\Delta_{\mathbf{m}}(u)| \le \prod_{i \in \{1...d\}, m_i > 0} \frac{6(\sqrt{e} b)^{2c_i} c_i}{c_i^{c_i}} u_i^{2c_i}.$$

As $6c_i \leq 6^{c_i}$, we have $|\Delta_{\mathbf{m}}(u)| \leq \prod_{i \in \{1...d\}, m_i > 0} \frac{(\sqrt{6}eb)^{2c_i}}{c_i^{c_i}} u_i^{2c_i}$. Next, applying Lemma 3 gives

$$|\Delta_{\mathbf{m}}(u)| \leq \left( \prod_{i \in \{1...d\}, m_i > 0} \frac{(\sqrt{6}eb)^{2c_i}}{c_i^{c_i}} \right) \left( M^{2\|c\|_1} \|c\|_1^{-\|c\|_1} \prod_{i \in \{1...d\}, m_i > 0} c_i^{c_i} \right)$$

$$= (\sqrt{6}eb)^{2\|c\|_1} M^{2\|c\|_1} \|c\|_1^{-\|c\|_1}$$

$$= (6eb^2 M^2)^{\|c\|_1} \|c\|_1^{-\|c\|_1}.$$

Since $\|c\|_1 \geq \|m\|_1 \geq A$, we can bound the error term with

$$\sup_{\|u\| \leq M} \left| k(u) - \tilde{k}(u) \right| \leq \sum_{\mathbf{m} \in \mathbb{N}^d, \mathbf{1}^\top \mathbf{m} > A} \sup_{\|u\| \leq M} |\Delta_{\mathbf{m}}(u)|$$

$$\leq \sum_{\mathbf{m}: \|\mathbf{m}\|_1 > A} (6eb^2 M^2)^{\|c\|_1} \|c\|_1^{-\|c\|_1}$$

$$\leq \sum_{\mathbf{m}: \|\mathbf{m}\|_1 > A} \left( \frac{6eb^2 M^2}{A} \right)^{\|c\|_1}.$$

Now, since by assumption $A \geq 24eb^2 M^2$ and $\|c\|_1 \geq \|m\|_1$, it follows that $6eb^2 M^2 / A \leq 1$ and so we can upper-bound this sum with

$$\sup_{\|u\| \leq M} \left| k(u) - \tilde{k}(u) \right| \leq \sum_{\mathbf{m}: \|\mathbf{m}\|_1 > A} \left( \frac{6eb^2 M^2}{A} \right)^{\|\mathbf{m}\|_1}$$

$$= \sum_{l=A+1}^{\infty} \sum_{\mathbf{m}: \|\mathbf{m}\|_1 = l} \left( \frac{6eb^2 M^2}{A} \right)^l$$

$$= \sum_{l=A+1}^{\infty} \binom{d+l-1}{l} \left( \frac{6eb^2 M^2}{A} \right)^l$$

$$\leq \sum_{l=A+1}^{\infty} 2^{d+l-1} \left( \frac{6eb^2 M^2}{A} \right)^l$$

$$= 2^{d-1} \sum_{l=A+1}^{\infty} \left( \frac{12eb^2 M^2}{A} \right)^l.$$

Summing this geometric series results in

$$\sup_{\|u\| \leq M} \left| k(u) - \tilde{k}(u) \right| \leq 2^{d-1} \left( \frac{12eb^2 M^2}{A} \right)^{A+1} \left( 1 - \frac{12eb^2 M^2}{A} \right)^{-1}$$

$$\leq 2^{d-1} \left( \frac{12eb^2 M^2}{A} \right)^A \left( 1 - \frac{1}{2} \right)^{-1}$$

$$= 2^d \left( \frac{12eb^2 M^2}{A} \right)^A.$$

This is what we wanted to show. $\qquad\square$

Using this, we can directly prove Corollary 3.

*Proof of Corollary 3.* Recall that the number of samples required for a sparse grid rule up to order $A$ is

$$D \leq 3^A \binom{d+A}{A} \leq 2^d \cdot 6^A \leq 2^d \exp(2A).$$

In order to ensure
$$\sup_{\|u\| \leq M} \left| k(u) - \tilde{k}(u) \right| \leq \epsilon,$$
it suffices by the result of Theorem 3 to have $A$ large enough that
$$2^d \left( \frac{12eb^2 M^2}{A} \right)^A \leq \epsilon$$
and
$$\frac{12eb^2 M^2}{A} < 1.$$
Suppose that we set $A$ such that
$$\frac{12eb^2 M^2}{A} \leq \exp(-2\gamma).$$
If $\gamma > 0$, then the second condition will be trivially satisfied. The first condition will also be satisfied when we set $A$ large enough that
$$2^d \exp(-2\gamma A) \leq \epsilon.$$
This occurs when
$$\exp(2A) \geq 2^{d/\gamma} \cdot \epsilon^{-1/\gamma}.$$
For this condition to hold, as $D \leq 2^d \exp(2A)$, it suffices to have
$$D \geq 2^d \cdot 2^{d/\gamma} \cdot \epsilon^{-1/\gamma}$$
samples. On the other hand, for $A$ to satisfy our original condition, we also need
$$A \geq 12eb^2 M^2 \exp(2\gamma).$$
This can be achieved when
$$D \geq 2^d \exp(24eb^2 M^2 \exp(2\gamma)).$$
Combining these two conditions using a maximum proves the corollary. $\qquad\square$

We now prove the technical lemma that we have used.

**Lemma 3.** *For any $u \in \mathbb{R}^d$ that satisfies $\|u\| \leq M$, and any $c \in \mathbb{R}^d$ with $c_i > 0$,*
$$\prod_{i=1}^d u_i^{2c_i} \leq M^{2\|c\|_1} \|c\|_1^{-\|c\|_1} \prod_{i=1}^d c_i^{c_i}.$$

*Proof.* We produce this result by optimizing over $u_i$. First, we let $x_i = u_i^2$, and note that an upper bound is
$$\prod u_i^{2c_i} \leq \max_{\sum x_i = M^2} \prod_{i=1}^d x_i^{c_i}.$$
Taking the logarithm and using the method of Lagrange multipliers to handle the constraint, we get Lagrangian
$$J(x, u) = \sum_{i=1}^d c_i \log(x_i) + u \left( M^2 - \sum_{i=1}^d x_i \right).$$
Differentiating to minimize gets us, for all $i$,
$$0 = \frac{c_i}{x_i} - u.$$
which results in
$$x_i = \frac{c_i}{u}.$$
In order to satisfy the constraint, we must set $u$ such that
$$x_i = \frac{c_i M^2}{\sum_{j=1}^d c_i} = \frac{c_i M^2}{\|c\|_1}.$$
With this assignment, we have
$$\prod u_i^{2c_i} \leq \prod_{i=1}^d \left( \frac{c_i M^2}{\|c\|_1} \right)^{c_i},$$
and simplification produces the desired result. $\qquad\square$

### C.3 Proof of Theorem 4

*Proof.* We first bound the approximation error of each sub-kernel acting on a subset $S$ of indices. Let $x_S, y_S$ be the vector $x, y$ restricted to these indices and let $k_S(x_S - y_S)$ be the sub-kernel acting on indices in $S$. As shown at the end of the proof of Theorem 2,

$$\left| k_S(x_S - y_S) - \tilde{k}_S(x_S - y_S) \right| \leq 3 \left( \frac{eb^2 \|x_S - y_S\|^2}{R} \right)^{\frac{R}{2}} = 3 \left( \frac{eb^2}{R} \right)^{\frac{R}{2}} \|x_S - y_S\|^R.$$

As $\mathcal{M}$ has diameter $M$, $\|x_S - y_S\| \leq M$. Noting that $R \geq 2$ since it is even, we can bound $\|x_S - y_S\|^R \leq \frac{\|x_S - y_S\|^2}{M^2} M^R$, and so

$$\left| k_S(x_S - y_S) - \tilde{k}_S(x_S - y_S) \right| \leq 3 \frac{\|x_S - y_S\|^2}{M^2} \left( \frac{eb^2 M^2}{R} \right)^{\frac{R}{2}}.$$

Summing over all $m$ sub-kernels, noting that each index only appears in at most $\Delta$ sets $S$, we have that $\sum_{S \in \mathcal{S}} \|x_S - y_S\|^2 \leq \Delta \|x - y\|^2 \leq \Delta M^2$. Therefore

$$\epsilon = \left| k(x - y) - \tilde{k}(x - y) \right| \leq 3\Delta \left( \frac{eb^2 M^2}{R} \right)^{R/2}.$$

The number of points we will use in total is $D \leq m\beta \binom{r+R}{r}$. By a similar argument as in Corollary 1, for any $\gamma > 0$, we obtain

$$D(\epsilon) \leq \beta m 2^r \max\left( \exp(e^{2\gamma+1}b^2 M^2), (3\Delta)^{1/\gamma} \epsilon^{-1/\gamma} \right).$$

$\square$

## D   Details of experiments

For the MNIST dataset, we use a 60k/10k split of train and test set. We use linear SVM on top of the features generated by random Fourier features or subsampled dense grid. The kernel bandwidth and the SVM hyper-parameters are chosen by cross validation.

For the task of acoustic modeling on the TIMIT dataset, the input features correspond to a frame of 10ms of speech, preprocessed using the standard feature pace Maximum Likelihood Linear Regression (fMMLR). The input dimension is 40. We use multinomial logistic regression on top of the features generated by random Fourier features, QMC, subsampled dense grid, or reweighted dense grid. The output of the multinomial logistic regressions is a probability distribution over 1917 groups of tri-phonemes. We constrain the weight matrix of the logistic regression to have rank at most 500, similar to [22]. The kernel bandwidth is chosen by performance on the validation set.