[Reviews · NeurIPS 2017]

Reviewer 1



%%% UPDATE: Thank you for your response, which has been read %%% %%% I had not appreciated that the experiments don't exactly implement the method for which theoretical results are provided; hence the reduction in my overall score by one level %%% This was an excellent paper - very careful, well written, impressive and enjoyable. It establishes solid theoretical results for deterministic quadrature alternatives to random Fourier features. I have a few comments, but overall I think this would be a fantastic contribution to NIPS. - On page 2, lines 72-74, I wonder if the assessment of QMC is a bit pessimistic. Was higher-order QMC used in that reference? Is there some fundamental reason to think that no QMC method at all would outperform the proposed approach? I am wondering about whether higher-order digital nets could be used within QMC; these correspond to weighted RKHS and have something in common with the ANOVA construction. See Dick and Pillichshammer's 2010 book. - \top for transpose in latex - define the Cartesian basis e_i before it is used - On page 4, lines 151-155, the comparison to RFF could be tempered again with a caveat that more assumptions are made to get the novel result, relative to the rate presented for RFF. Put another way, is it clear that RFF would not perform better under this additional assumption being made in the paper? - Please define \tilde{\Omega} in section 4.

Reviewer 2



Post-rebuttal comments: Thank you for the feedback. I have raised my score on overall rating. Summary: In my opinion, this is an interesting and promising direction for the use of deterministic quadrature rules in kernel approximation. The weakness of the paper is in the experiments: there should be more complete comparisons in computation time, and comparisons with QMC-based methods of Yang et al (ICML2014). Without this the advantage of the proposed method remains unclear. - The limitation of the obtained results: The authors assume that the spectrum of a kernel is sub-gaussian. This is OK, as the popular Gaussian kernels are in this class. However, another popular class of kernels such as Matern kernels are not included, since their spectrum only decay polynomially. In this sense, the results of the paper could be restrictive. - Eq. (3): What is $e_l$? Corollaries 1, 2 and 3 and Theorem 4: All of these results have exponential dependence on the diameter $M$ of the domain of data: a required feature size increases exponentially as $M$ grows. While this factor does not increase as a required amount of error $\varepsilon$ decreases, the dependence on $M$ affects the constant factor of the required feature size. In fact, Figure 1 shows that the performance is more quickly getting worse than standard random features. This may exhibit the weakness of the proposed approaches (or at least of the theoretical results). - The equation in Line 170: What is $e_i$? - Subsampled dense grid: This approach is what the authors used in Section 5 on experiments. However, it looks that there is no theoretical guarantee for this method. Those having theoretical guarantees seem not to be practically useful. - Reweighted grid quadrature: (i) It looks that there is no theoretical guarantee with this method. (ii) The approach reminds me of Bayesian quadrature, which essentially obtains the weights by minimizing the worst case error in the unit ball of an RKHS. I would like to look at comparison with this approach. (iii) Would it be possible to derive a time complexity? (iv) How do you chose the regularization parameter $\lambda$ in the case of the $\ell_1$ approach? - Experiments in Section 5: (i) The authors reported the results of computation time very briefly (320 secs vs. 384 seconds for 28800 features in MNIST and "The quadrature-based features ... are about twice as fast to generate, compared to random Fourier features ..." in TIMIT). I do not they are not enough: the authors should report the results in the form of Tables, for example, varying the number of features. (ii) There should be comparison with the QMC-based methods of Yang et al. (ICML2014, JMLR2016). It is not clear what is the advantage of the proposed method over the QMC-based methods. (iii) There should be explanation on the settings of the MNIST and TIMIT classification tasks: what classifiers did you use, and how did you determine the hyper-parameters of these methods? At least such explantion should be included in the appendix.

Reviewer 3



I have read the other reviews and the rebuttal. This is paper is concerned with deterministically constructing a feature map z for the class of subgaussian kernels, which can serve as a substitute for random Fourier transforms. By Bochner's theorem, expressing a kernel via its Fourier transforms amounts to computing an intractable integral. While, classically, this integral is computed by MC integration, the authors propose to use Gaussian quadrature to calculate this Fourier transform. For the special case of the ANOVA kernel, the Gaussian quadrature based Fourier transform is shown to produce good estimates using O(D) instead of O(D^3). The good performance of this method is backed up by experiments for classification problems on MNIST and TIMIT. The paper reads very well. The introduction on kernels and quadrature is well-written and serves as a good introduction. I liked the fact that the involved proofs are in the appendix, however I would recommend to mention this in the main text, since I at first thought that they were simply omitted. This research is highly significant. Substituting high-dimensional Monte Carlo integration with deterministic ones can---as the paper shows---yield significant speed-ups, if it is done right. Of course this will only work, if one fits the integration rule to the integrand. This case, where the ANOVA kernel is used, is a convincing example, since it encodes sparse dependencies and therefore the 'curse of dimensionality', i.e. the room for a function to wiggle around uncontrollably in unmodelled dimensions, can be avoided. I think this would be very interesting for the community to see. As an addition I would like to point the authors to an interesting line of research concerning probabilistic versions of Gaussian quadrature rules, about which the authors might not know. Sarkka et al [1] have shown that one can obtain Gaussian quadrature rules (i.e. polynomially-exact quadrature) as a posterior mean for Bayesian quadrature with suitable kernels. Maybe this can be combined with your methods, if the numerical uncertainty over the integration for the Fourier map is important? As a last point of criticism, I want to point out that one of the fundamental references in this paper---[18] "How to cale up kernel methods to be as good as neural nets"---is only an arxiv submission and appears not to be peer-reviewed although it was published in 2014. Is this the case or was the wrong reference chosen? For a reference which is cited as a fundamental reason for this paper I would expect something which is peer-reviewed. Overall, I think it's a very good paper and due to its interesting theoretical and practical implications I vote to 'accept'. [1] https://arxiv.org/pdf/1504.05994.pdf